# Construction of Charring-Functional Polyheptanazine towards Improvements in Flame Retardants of Polyurethane

**DOI:** 10.3390/molecules26020340

**Published:** 2021-01-11

**Authors:** Shaolin Lu, Botao Shen, Xudong Chen

**Affiliations:** Key Laboratory for Polymeric Composite and Functional Materials of Ministry of Education, School of Chemistry, Sun Yat-sen University, Guangzhou 510275, China; lushlin@mail2.sysu.edu.cn (S.L.); shenbt@mail2.sysu.edu.cn (B.S.)

**Keywords:** flame retardants, thermoplastic polyurethane, phosphorous doped, polyheptanazine, thermal decomposition

## Abstract

Nitrogen-containing flame retardants have been extensively applied due to their low toxicity and smoke-suppression properties; however, their poor charring ability restricts their applications. Herein, a representative nitrogen-containing flame retardant, polyheptanazine, was investigated. Two novel, cost-effective phosphorus-doped polyheptazine (PCN) and cobalt-anchored PCN (Co@PCN) flame retardants were synthesized via a thermal condensation method. The X-ray photoelectron spectroscopy (XPS) results indicated effective doping of P into triazine. Then, flame-retardant particles were introduced into thermoplastic polyurethane (TPU) using a melt-blending approach. The introduction of 3 wt% PCN and Co@PCN could remarkably suppress peak heat release rate (pHRR) (48.5% and 40.0%), peak smoke production rate (pSPR) (25.5% and 21.8%), and increasing residues (10.18 wt%→17.04 wt% and 14.08 wt%). Improvements in charring stability and flame retardancy were ascribed to the formation of P–N bonds and P=N bonds in triazine rings, which promoted the retention of P in the condensed phase, which produced additional high-quality residues.

## 1. Introduction

Nitrogen-containing compounds are used as halogen-free flame retardants for polymers due to their low toxicity and smoke suppression during fires [1]. Nitrogen-containing compounds improve flame retardancy mainly by diluting the concentration of oxygen and by encouraging the dripping of a polymer [2,3]. However, the poor flame-retardant efficiency of nitrogen-containing flame retardants restricts their applications. Inspired by the condensed-phase flame retardancy mechanism, the formation of more char residues in the condensed phase is a superior way to reduce the heat release rate (HRR) during combustion.

Triazine-based compounds are halogen-free flame retardants frequently used for thermoplastic polyurethane (TPU) because of their smoke suppression during combustion and their low toxicity [4,5]. Most triazine-based flame retardants decompose to produce noncombustible gases, which act as gas-phase flame retardants [6]. Polyheptazines, derivatives of triazine flame retardants, promote a condensed-phase flame retardancy mechanism by forming a physical barrier in a polymer matrix [7,8,9]. All of these flame-retardant systems have demonstrated that no reaction occurs between flame retardants and the polymer matrix during degradation, which does not change the residue production. Thus, it is important to investigate methods to realize charring of the condensed phase without changing the two-dimensional framework of polyheptanazine. Recent studies have demonstrated that phosphorus doping can improve the thermo-oxidative stability of graphene oxide (GO) and reduced graphene oxide (rGO) [10]. Polyheptanazine contains nitrogen triangles with six lone-pairs of electrons, which are available for doping [11]. Phosphorus can be used in flame retardants and may replace the halogenated variants currently in use [12]. Research on P-containing flame retardants is increasing due to environmental pollution caused by halogenated flame retardants [13,14]. A variety of halogen-free flame retardants for polymers have been developed, most of which contain P [15,16,17,18]. In recent years, phosphorus-doped polyheptazines have attracted increasing attention due to their applications in wastewater treatment [19], photocatalysts [20,21], electrocatalysts [22], organic matter degradation [23,24], and bioimaging [25]. According to the literature [12], compounds containing P–N bonds can promote the formation of more stable char residues due to the formation of a crosslinked network between P–N bond compounds and polymer chains during combustion. Therefore, it is speculated that P–N bonds can be formed by controlling interstitial P doping to obtain more effective polyheptazine flame retardants.

Transition metals can also improve the flame retardancy of TPU [26,27]. Shi et al. [28] prepared spinel copper cobaltate/polyheptazine nanohybrids (C–CuCo_2_O_4_) via a hydrothermal method. The results showed that the introduction of C–CuCo_2_O_4_ into TPU achieved a flame-retardant effect and reduced the release of toxic gases. In his other work, polystyrene/g-C_3_N_4_/ZnCo_2_O_4_ composites were prepared [29], and the results showed that the introduction of hybrids enhanced the thermal stability of PS during catalysis. The introduction of a Co compound into the polymer can reduce the evolution of aromatic compounds, which means the reduction of toxic organic volatiles [30,31]. Many studies have shown that Co can improve the charring property of polymers and can suppress the release of toxic gases during degradation [32]. It has been reported that an abundance of N atoms on the surface of polyheptanazine provides a tremendous number of adsorption sites for anchoring metal ions [22]. Generally, analogous to Fe–N bonds [33], the specific interactions between Co–N bonds is useful for improving water oxidation [34].

To enhance the effect of condensed-phase flame retardancy of polyheptazine (CN), a phosphorus-doped polyheptanazine (PCN) was prepared by using the diammonium hydrogen phosphate as the P source and urea as the polyheptazine precursor. After further heat treatment with cobalt acetate, Co was successfully anchored on the surface of phosphorus-doped polyheptazine (Co@PCN). Afterward, these flame-retardant particles were incorporated into TPU, and then the thermal stability and fire hazards of the nanocomposites were investigated. Introducing P and Co is expected to improve the poor condensed-phase flame retardancy typically observed in nitrogen-containing flame retardants.

## 2. Results and Discussion

### 2.1. Characterization of PCN and Co@PCN

As shown in Figure 1a, Fourier-transform infrared (FTIR) spectroscopy was used to define the functional groups and chemical species in the prepared particles. In general, the original CN contains three typical vibration bands, included N–H and C–N bonds, located around 3200 cm^−1^ and 1200–1650 cm^−1^, respectively. The third peak was detected at 814 cm^−1^, which was caused by the out-of-plane bending vibration of heptanazine rings [35]. The vibrational bands of P–N functional groups of PCN and Co@PCN were not detected, which may be due to the overlap of these vibrational bands with strong C–N vibrational bands [36,37,38]. Nevertheless, a small wavenumber shift occurred in the sharp band around 814 cm^−1^ (Figure 1b), indicating a change in the electron cloud in C–N bonds and C=N bonds induced by phosphorus doping [39]. A similar phenomenon was also observed in Co@PCN. There was also a new band at 495 cm^−1^ in PCN and Co@PCN, which originated from the P–O–P bending mode [40]. The crystal structure and phase purity of PCN and Co@PCN were assessed by X-ray diffraction (XRD) using CN as a reference sample. The results are shown in Figure 1c. The pattern of CN had two characteristic peaks. The peak at 12.97° was caused by in-plane repeating units at, and the other one at 27.73° originated from the interlayer stacking of conjugated aromatic units [35]. The PCN particles showed a similar XRD pattern to the original polyheptanazine, which indicates that polyheptanazine retained its framework well during the phosphorization process. However, there were some small sharp peaks in the P-doped samples, indicating that the samples may contain crystalline phosphate impurities [23,40,41]. In addition, the weakening of the PCN (002) diffraction peak may be caused by the inclusion of hetero-element P during thermal phosphorization, which enlarged the interlayer spacing and reduced the thickness of the graphitic structure. The further decrease in the diffraction intensity of metal-containing samples (Co@PCN) may be attributed to chemical coupling between PCN and metal phosphide clusters [35,42]. Figure 1d shows the TG-DSC curve of a mixture of urea and diammonium hydrogen phosphate (DHP). The endothermic peak of the mixture at 134 °C was the melting of urea. The melting point of DHP is about 155 °C. Figure 1d shows that urea and DHP began to react at 150–230 °C. The corresponding peak at 237 °C was due to the formation of a triazine structure. The polymerization of the triazine compound occurred at 364 °C, which was earlier than the similar peak temperature of CN prepared using pristine urea [3]. The residues of the mixture of urea and DHP were 3.8% at 550 °C from the curves of TG-DSC (Simultaneous thermogravimetric analysis coupled with differential scanning calorimeter) The obtained product yield (PCN) was about 5.2% from calcination in muffle furnace. The reason for the differences may be due to the different environments in which mixture were located.

As shown in Figure 2, similar to that of CN and PCN, Co@PCN still presented a distinct nanosheet structure. The composition of chemical species in Co@PCN particles was further investigated using scanning transmission electron microscope (STEM) and energy-dispersive spectrum mapping (EDS) analysis. Significantly, P and Co showed analogous distribution patterns as N (Figure 2h,i), which demonstrated effective doping of P into the triazine rings of CN. We did not observe nanoclusters on the surface of Co@PCN. It should be noted that the low-density distribution of Co suggested that there were other forms of CoP or cobalt oxides besides the nanoclusters. A considerable part of these may be only sub-nanometer or even atomic in size [35].

The X-ray photoelectron spectroscopy (XPS) analysis images of CN, PCN, and Co@PCN are given in Figure 3, and the atomic percentages of these particles are listed in Table 1. In the C 1s spectrum (Figure 3b) of CN, the peaks at 285.08 and 288.28 eV were attributed to the sp^2^ graphitic carbon (C–C=C) and sp^2^-bonded carbon in the triazine rings (N–C=N), respectively [43]. CN has a lower area ratio of graphitic carbon than PCN, and C–O (289.48 eV) appeared. The C 1s XPS spectra of Co@PCN can be divided into four peaks at 284.93, 287.23, 288.13, and 288.78 eV, corresponding to sp^2^ graphitic C, C–N, sp^2^-bonded carbon in N–C=N, and sp^2^-hybridized C bonded to –NH_2_ on the aromatic ring, respectively [22]. The XPS N 1s spectra of CN (Figure 3c) at 398.68, 399.28, and 400.68 eV were attributed to sp^2^-hybridized nitrogen C–N=C, tertiary nitrogen N–C3, and C–N–H (positive charge localization in heterocycles), respectively [35]. The last peak positioned at 401.78 eV was identified as C–NH_2_ in PCN [44,45]. Two additional peaks for Co@PCN located at 398.98 and 400.48 eV corresponded to N–Co and C–NH_2_, respectively [34,46]. The C 1s XPS spectra of PCN and Co@PCN (Figure 3d) can be divided into two peaks at 531.43 eV and 532.88 eV, corresponding to adsorbed O and oxygen and H_2_O on the surface [22]. The P 2p XPS spectra of PCN and Co@PCN contained peaks at 133.13, 133.93, and 134.83 eV, which correspond to P–N, P=N, and P–O bonds, respectively (Figure 3e) [36,39,47]. This implied that P likely replaced C in the *s*-triazine units to form P–N and P=N bonds because the binding energies of P–C coordination bonds (131.2–132.2 eV) were lower than that of P–N and P=N bonds [22]. P–O bonds were ascribed to oxidized P on the surface [39]. The Co 2p spectrum is displayed in Figure 3f, and the peaks positioned at 781.43 and 797.63 eV were associated with Co^3+^. The peaks at 783.03 and 801.48 eV were related to Co^2+^ [48,49]. The peaks at 803.98 and 786.48 eV were assigned to the satellite peaks of Co 2p_1/2_ and 2p_3/2_ [49]. The peaks at 779.6 and 795.0 eV were related to the Co−N_x_ structure of Co@PCN, which formed due to van der Waals forces [50,51].

The thermogravimetric analysis (TGA) curves of CN, PCN, and Co@PCN are shown in Figure 4 (in nitrogen), and the corresponding thermal decomposition data are given in Table 2. The initial decomposition temperature, the temperature at 50% weight loss, and the temperature at the maximal weight were recorded as T_−5_, T_−__50_, and T_−*max*_, respectively. CN presented a one-step decomposition process over the whole temperature range. Unlike CN, PCN exhibited three different thermal decomposition processes (Figure 4b). The decomposition of PCN at 200–550 °C may be attributed to the decomposition of a small amount of oxidized P and oligomers. The T_−50_ of PCN and Co@PCN were higher than that of CN, indicating that the formation of a graphitic polyheptanazine framework was not affected by phosphorus doping or cobalt loading. The residues of PCN and Co@PCN at 750 °C were 26.3% and 47.1%, respectively, which were much higher than that of CN, typically 0% at 750 °C. This indicated that phosphorus doping greatly increased the residues of CN, possibly due to the formation of P−N bonds, which encouraged the retention of P in the condensed phase during thermal decomposition.

### 2.2. Fracture Surface Analysis of TPU and its Nanocomposites

Dispersions of CN, PCN, and Co@PCN with the TPU matrix were studied using SEM. From Figure 5a, pure TPU had a relatively smooth and fractured surface while the fracture surfaces of CN-TPU, PCN-TPU, and Co@PCN-TPU were rough and uneven. CN particles were tightly embedded in TPU (Figure 5b), demonstrating their strong interactions with the polymer. As shown in Figure 5c, aggregations were observed in PCN-TPU, suggesting the poor dispersion of PCN particles in the TPU matrix. For Co@PCN-TPU (Figure 5d), the aggregates dispersed in the TPU matrix were larger than those in PCN-TPU. This may lead to poor flame retardancy test performance. SEM images of the fracture surface of PCN-TPU and Co@PCN-TPU at different rotating speeds were also investigated (Appendix A). It can be concluded that changing the rotational speed has a limited effect on improving the dispersion of flame-retardant additives. CN particles were well-dispersed, as shown by analysis of the fracture surface of TPU and its nanocomposites. However, the condensed-phase flame retardancy mechanism of CN did not promote the formation of residues and the flame retardancy effect was still much lower than that of PCN and Co@PCN.

### 2.3. Thermal Stability and Fire Resistance of TPU and Its Composites

The thermal decomposition of the TPU composites in nitrogen is shown in Figure 6, and the corresponding parameters are listed in Table 3. All nanocomposites exhibited similar degradation behavior to that of pure TPU (Figure 6). T_−5_ of CN-TPU decreased slightly after the introduction of CN, while T_−5_ of PCN-TPU was 19 °C lower than that of pure TPU, which may be attributed to the presence of a small amount of oxidized P in PCN. The initial decomposition temperature of Co@PCN-TPU was 4 °C higher than that of pure TPU, which was due to the higher thermal stability of Co@PCN. T_−50_ of CN-TPU and PCN-TPU were basically the same as that of pure TPU, while T_−50_ of Co@PCN-TPU was 10 °C higher than that of pure TPU. The residues of CN-TPU, PCN-TPU, and Co@PCN-TPU at 700 °C were 5.2, 9.5, and 10.7%, respectively, which were 4.4% higher than pure TPU. This phenomenon demonstrated that the introduction of phosphorus increased the residues, which was attributed to the effect of P–N bonds [12]. The overall influence of PCN and Co@PCN was that the incorporated additives did not influence the decomposition of polyurethane bonds but slightly increased the residue percentage formed during the decomposition of polyols (the second decomposition step in Figure 5b).

The flame retardancy of TPU and its nanocomposites under nitrogen atmosphere was evaluated by microcalorimetry (MCC). As shown in Figure 7 and Table 4, the peak heat release rate (pHRR) and total heat release (THR) of pure TPU during combustion were 407.8 kW/m^2^ and 86.34 MJ/m^2^, respectively. The pHRR for PCN-TPU was 33.8% lower while the pHRR of Co@PCN-TPU decreased by 16.4% compared with pure TPU. It can be seen that CN has a good thermal inhibition effect in a nitrogen atmosphere. Figure 7b shows that the incorporation of PCN and Co@PCN further reduced the THR of the nanocomposites. It demonstrated that the introduction of phosphorus or cobalt is beneficial to further reduce the THR of nanocomposites under nitrogen atmosphere.

The HRR and THR curves obtained from cone calorimetry tests (CCT) of pure TPU and its nanocomposites are shown in Figure 8. The pHRR of pure TPU was 509.4 kW/m^2^ while that of CN-TPU was 561.4 kW/m^2^, which was the highest of all samples (Figure 8a). It demonstrated that the addition of untreated CN alone did not help reduce the pHRR of thermoplastic polyurethane. After the introduction of PCN and Co@PCN into the polymer, the pHRR decreased by 48.5% and 40.1% for PCN-TPU and Co@PCN-TPU, respectively, rivaled by pure TPU. PCN and Co@PCN changed the pyrolysis process of TPU, forming more char layers, which was mainly attributed to the formation of P–N bonds and P=N bonds in triazine rings, which promoted the retention of P in the condensed phase and produced more high-quality residues.

The results in Figure 8b show that PCN-TPU and Co@PCN-TPU degraded slightly earlier than pure TPU samples. The THR of pure TPU, CN-TPU, PCN-TPU, and Co@PCN-TPU were 86.34 MJ/m^2^, 82.31 MJ/m^2^, 74.01 MJ/m^2^, and 76.68 MJ/m^2^, respectively (Table 4). PCN and Co@PCN encouraged the production of more residues and reduced the mass loss, thus reducing the value of total heat release. The CCT results showed that the flame-retardant efficiency of PCN was better than that of Co@PCN.

The smoke-production rate (SPR) curves of pure TPU and its composites are shown in Figure 8c. The pSPR was greatly reduced after the introduction of PCN and Co@PCN, indicating that smoke toxicity of TPU decreased. Under the same filling contents, PCN-TPU presented the lowest pSPR and showed the best smoke-suppression performance of all samples. This was attributed to the charring ability of the condensed phase of modified polyheptanazine, which reduced heat release and smoke production.

Figure 8d gives the mass loss curves of nanocomposites as a function of time. The lower mass loss was ascribed to the formation of a high-quality char on the nanocomposite surface and its structure improvement [52]. Only 10.18% of pure TPU remained after the CCT test. The PCN-TPU plot shows a high residue value of 17.04%, which indicates that P-doped polyheptanazine was cross-linked. Moreover, the char residues formed on the surface can prevent heat and mass transfer, further reducing mass loss and improving fire safety.

It has been reported that the char residues of cyclo-P–N bond composites include P–O–P bonds and P–O–C bonds [53,54]. Combined with the TG data, it can be inferred that the increase of residues during cone calorimetry tests may be attributed to the production of phosphoric acid or polyphosphate during pyrolysis, which promotes the formation of heat-resistant carbonaceous compounds.

The curves of CO and CO_2_ evolution as a function of time obtained from CCT are shown in Figure 8e,f, and the total amount of CO and CO_2_, and the corresponding CO_2_/CO ratio of TPU and its nanocomposites after complete combustion are listed in Table 5. CO and CO_2_ are the main components of fire gas. A high concentration of CO will lead to carbon monoxide poisoning due to obstruction of escape in a fire. The lower values of CO_2_/CO ratio indicated that the low efficiency of combustion and the conversion of CO to CO_2_ are inhibited [55]. As shown in Table 5, the yields of CO of PCN-TPU and Co@PCN-TPU were higher than that of pure TPU, but PCN-TPU and Co@PCN-TPU had lower yields of CO_2_ compared to pure TPU, leading to a decrease in CO_2_/CO ratios (29.7 and 27.8 vs. 44.9). This may be due to the reaction between a small amount of oxidized P with the polymer matrix, which prevents further oxidation of CO release from PCN-TPU and Co@PCN-TPU in the first combustion process into CO_2_, whereas that reaction could not occur in the combustion process of pure TPU.

TPU and its nanocomposites passed the UL-94 V-2 rating with melt dripping in the vertical burning test. Furthermore, the dripping object ignited the absorbent cotton. t1 + t2 (Total burning time is used to evaluate the UL-94 rating.) in the UL-94 test decreased as the flame retardants were incorporated. t1 + t2 of pure TPU is 3 s while those of PCN-TPU and Co@PCN-TPU are shorter.

To highlight the improved fire performance of phosphorus-doped polyheptanazine in the TPU matrix, the performance of PCN-TPU was compared with the reported results of other flame-retardant TPU nanocomposites (Table 6). It can be inferred that PCN encourages the formation of char residues, resulting in decreased pHRR values. Table 6 also shows that PCN has a definite smoke-suppression effect. In this work, the introduction of PCN to TPU remarkably reduced the PHRR (48.5%), which was better than most reported results. Yang et al. [56] fabricated cetyltrimethylammonium bromide (CTAB)-modified Ti_3_C_2_ (MXene) ultra-thin nanosheets (CTAB-Ti_3_C_2_). Despite excellent flame retardance properties (−51.2% in pHRR), the use of MXene flame retardants has been greatly limited because their preparation methods are expensive and difficult to industrialize. The raw materials used in this paper are urea and diammonium hydrogen phosphate, which are produced industrially, making them easy to acquire. In addition, the preparation method of PCN is simple.

### 2.4. Evolution of Pyrolysis Gas

Simultaneous thermal analysis coupled with Fourier-transform infrared spectrometry (TG-IR) was used to detect the effect of those particles on the pyrolysis behavior of pure TPU, CN-TPU, PCN-TPU, and Co@PCN-TPU (Figure 9). TPU nanocomposites showed similar main spectra bands to that of pure TPU, and those bands were attributed to functional groups or components with characteristic band positions (Figure 10), including CO_2_ (2358 cm^−1^), aromatic compounds (1508 and 1460 cm^−1^), esters (1145 cm^−1^), hydrocarbons (2977 and 2880 cm^−1^), carbonyl compounds (1751 cm^−1^), and HCN (714 cm^−1^) [57,62,63]. The results showed that pure TPU and CN-TPU exhibited similar thermal decomposition processes while PCN-TPU decomposed slightly earlier. This demonstrated that incorporating these flame-retardant particles into TPU may not change the primary decomposition process.

Figure 10a shows that the decomposition of PCN-TPU released abundant CO_2_ during the first stage, which was related to the slightly earlier decomposition due to the addition of PCN during TGA (Figure 6) and MCC (Figure 7). This can enhance the gas-phase flame-retardant mechanism of the nanocomposites. The inhibition of PCN-TPU was better than that of Co@ PCN-TPU in the remaining characteristic band analysis. This can be attributed to the better char-formation ability of PCN in the matrix. Figure 10b shows the evolution trend of aromatic compounds. The addition of three fillers increased the intensity of the first stage and decreased the intensity of the second stage, which did not prolong the ignition time of the nanocomposite during CCT. This is different from the previous literature that stated Co compounds can inhibit the formation of aromatic compounds [31]. In addition, PCN-TPU showed a different phenomenon from other samples in the hydrocarbon curve in Figure 10d. A few hydrocarbons were released during the first stage, which may be attributed to the reaction of a small amount of oxidized P with the polymer matrix, corresponding to premature decomposition of PCN-TPU in TGA. Figure 10f shows that PCN changed the decomposition pathway, thus avoiding HCN produced by polyheptanazine decomposition. This stage corresponds to the 500–600 °C decomposition stage in TGA.

## 3. Materials and Methods

### 3.1. Materials

TPU (S80A, BASF SE) with a density of 1.21 g/cm^3^, urea (99%, J&K Scientific, Ltd., Beijing, China), diammonium hydrogen phosphate (99%, Shanghai Macklin Biochemical Co., Ltd., Shanghai, China), Cobalt (II) acetate tetrahydrate (99.5%, Shanghai Macklin Biochemical Co., Ltd.), and ethanol (≥99.7%, Shanghai Titan Scientific Co., Ltd., Shanghai, China) were used as received without further purification.

### 3.2. Preparation of Phosphorus-Doped Polyheptazine Nanomaterials (PCN)

A preparation scheme of PCN and Co@PCN is shown in Scheme 1. PCN were synthesized by the thermal-condensation method using diammonium hydrogen phosphate as the P source and urea as the polyheptazine precursor. In detail, 40 g of urea and 2.4 g of diammonium hydrogen phosphate were thoroughly mixed, and the resulting mixture was placed in an open porcelain crucible. The crucible was placed in a muffle furnace (KSL-1200X, Hefei Ke Jing Materials Technology Co., Ltd., Hefei, China) and calcined at 550 °C for 2 h with a heating rate of 5 °C min^−1^. The obtained yellow product was crushed with a grinder into fine grains for further use. In the absence of diammonium hydrogen phosphate, CN was synthesized by urea using the same steps.

### 3.3. Preparation of Co@PCN Nanomaterials

Cobalt (II) acetate tetrahydrate (0.9 g) and 3.6 g of PCN were dispersed in ethanol, and then, ethanol was evaporated at 50 °C under magnetic stirring. After drying at 60 °C for 12 h, the mixture was heated at a rate of 5 °C min^−1^ in a muffle furnace from room temperature to 500 °C and maintained for 3 h. Finally, a wathet blue product was obtained.

### 3.4. Preparation of Flame-Retarding TPU

Pristine TPU and its composites were fabricated by a melt blending method. Typically, 1.5 g of Co@PCN was blended with 48.5 g of TPU at 180 °C in a torque rheometer (RHEOCORD 300P, Germany) for 15 min with a rotor speed of 80 rpm. Next, the sample containing 3.0 wt% Co@PCN was hot-pressed via a flat vulcanizing machine at 185 °C to obtain sheets with suitable sizes, which were named TPU/Co@PCN. Other samples with 3.0 wt% CN and 3.0 wt% PCN were labeled CN-TPU and PCN-TPU, respectively. The control sample of pure TPU was also prepared by the same process.

### 3.5. Characterization Procedures

Fourier-transform infrared (FT-IR) spectra were obtained on an FTIR spectrometer (EQUINOX 55, Bruker, Ettlingen, Germany) with scanning from 4000 cm^−1^ to 400 cm^−1^.

X-ray diffraction (XRD) patterns were measured by using an X-ray diffractometer (Empyrean, Malvern PANalytical, Almelo, The Netherlands) equipped with CuKα radiation.

Simultaneous thermogravimetric analysis coupled with differential scanning calorimeter (TG-DSC) was carried out on a simultaneous thermal analyzer (STA449F3/Nicolet6700).

Transmission electron microscopy (TEM) images and energy-dispersive spectroscopy (EDS) analyses were obtained using a FEI Tecnai G2 Spirit instrument with an acceleration voltage of 200 kV.

The binding energies of C, N, O, P, and Co in the samples were determined by X-ray photoelectron spectroscopy (XPS) on a Thermo Fisher Scientific system equipped with Al Kα radiation (hν = 1486.8 eV).

Thermogravimetric analysis (TGA) was carried out on a NETZSCH TG209F1 Libra thermogravimetric analyzer.

The morphology of the prepared composites was investigated by scanning electron microscopy (SEM). The samples were mounted onto an aluminum stub, sputter-coated with gold, and imaged by SEM (Hitachi S-4800 field-emission microscope, Tokyo, Japan) at an accelerating voltage of 10 kV. The fracture surface of the flame-retardant samples was prepared by quenching in liquid nitrogen.

Small-scale combustion performance was investigated by a Govmark MCC-2 microcalorimeter (New York, NY, USA). Specifically, 5 mg of samples were heated to 750 °C with a ramp rate of 1 °C/s in a nitrogen stream flowing at 80 mL/min. The pyrolysis products were mixed with oxygen (20 mL/min) and then burned in a combustion furnace at 900 °C.

Cone calorimetry tests (CCT) were carried out by an FTT0007 cone calorimeter with 100 × 100 × 3.5 mm^3^. The radiation heat flux was 35 kW/m^3^.

UL-94 vertical flame test was measured on a UL94-X type instrument (UL94 Flame Chamber, Motis). The specimens used for the test were of the dimensions 100 × 13 × 3.6 mm^3^.

Simultaneous thermal analysis was interfaced with Fourier-transform infrared spectrometry (TG-IR). The thermal analysis was performed from 30 to 800 °C with a heating rate of 10 °C min^−1^ under a nitrogen atmosphere.

## 4. Conclusions

In this work, PCN and Co@PCN, two functionalized polyheptanazines with enhanced charring, were prepared via thermal condensation and used as flame-retardant fillers for TPU. The results demonstrated that P replaced C in the *s*-triazine units to form P–N and P=N bonds. After ultrasonic treatment of Co@PCN, STEM images showed that Co still had a similar distribution pattern to that of N, indicating that Co was successfully anchored on the surface of doped polyheptanazine. Cone calorimetry tests showed that PCN-TPU and Co@PCN-TPU remarkably decreased the pHRR, THR, and pSPR compared with pure TPU. In particular, using only 3.0 wt% filling, the pHRR was decreased by 48.5% and 40.1%, respectively. The enhanced flame retardancy was mainly attributed to the formation of P–N and P=N bonds in triazine rings, which promoted the retention of P in the condensed phase and produced more high-quality residues. The PCN flame retardant suppressed the production of combustible and toxic gases to a certain extent. However, the additional Co anchored on the surface of doped polyheptanazine has no significant effect on the flame retardancy of polymer composites. Furthermore, Co@PCN did not suppress the production of toxic gases such as CO. The effect of in situ introduction of P and CO on the charring of polyheptylazine was studied. The results demonstrated that the flame-retardant effect of in situ introduction of P is better than that of Co. This work may provide a design route to solve the problem whereby nitrogen-containing flame retardants lack a condensed-phase charring effect.

## Data Availability

The data presented in this study are available in Appendix A.

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
