# Peer review of "Construction of Charring-Functional Polyheptanazine towards Improvements in Flame Retardants of Polyurethane"

_molecules, 2021, doi:10.3390/molecules26020340_

Round 1
Reviewer 1 Report
The work presented in this manuscript looks interesting and can be accepted for publication after considering the following minor corrections.
1. Abstract need to be re-written highighting what is new and achievements in numbers.
2. Introduction need citation of potentional references in this topic.
3. Comparative study is needed with reported papers and explaining why this materials is better in terms of efficiency ( Flame retardancy and toxic gases suppression) and cost.
4. English need improvement.
5. Conclusion need to be improved highlighting the new and significant findings.
Reviewer 2 Report
The manuscript corresponds to the Journal aims and scopes. This study seems to be interesting and the use of a variety of research techniques allows the conclusions to be considered as well-founded.
However, it can be accepted to publication after major revision in accordance to the remarks noted below.
- Based on the relatively rich experimental material, the authors should try to explain the mechanisms of the observed differences in the properties of composites more presciently. As the authors conclude, this would significantly increase the value of the article for defining a problem-solving strategy for lack of charring effect.
- The thesis put forward in chapter 2.4 on the basis of only SEM images of fracture surface, seems to be unreliable. In order to unequivocally confirm the effect of CN, PCN and Co @ PCN particle dispersion and aggregation in the TPU matrix, composites with a different dispersion of the flame-retardant additive, should also be tested. Such samples can be easily prepared by changing the blending parameters.
- The language of the article requires careful correction due to the large number of grammatical and spelling errors.
Round 2
Reviewer 2 Report
The authors made changes in line with the reviewer's suggestions. I believe that the article can be published as is.
Author Response
Thank you for your comment.